# Neuron Merging: Compensating for Pruned Neurons

**Woojeong Kim**      **Suhyun Kim**∗      **Mincheol Park**      **Geonseok Jeon**
Korea Institute of Science and Technology
{kwj962004, dr.suhyun.kim, lotsberry, hotchya}@gmail.com

## Abstract

Network pruning is widely used to lighten and accelerate neural network models. Structured network pruning discards the whole neuron or filter, leading to accuracy loss. In this work, we propose a novel concept of neuron merging applicable to both fully connected layers and convolution layers, which compensates for the information loss due to the pruned neurons/filters. Neuron merging starts with decomposing the original weights into two matrices/tensors. One of them becomes the new weights for the current layer, and the other is what we name a scaling matrix, guiding the combination of neurons. If the activation function is ReLU, the scaling matrix can be absorbed into the next layer under certain conditions, compensating for the removed neurons. We also propose a data-free and inexpensive method to decompose the weights by utilizing the cosine similarity between neurons. Compared to the pruned model with the same topology, our merged model better preserves the output feature map of the original model; thus, it maintains the accuracy after pruning without fine-tuning. We demonstrate the effectiveness of our approach over network pruning for various model architectures and datasets. As an example, for VGG-16 on CIFAR-10, we achieve an accuracy of 93.16% while reducing 64% of total parameters, without any fine-tuning. The code can be found here: https://github.com/friendshipkim/neuron-merging

## 1   Introduction

Modern Convolutional Neural Network (CNN) models have shown outstanding performance in many computer vision tasks. However, due to their numerous parameters and computation, it remains challenging to deploy them to mobile phones or edge devices. One of the widely used methods to lighten and accelerate the network is pruning. Network pruning exploits the findings that the network is highly over-parameterized. For example, Denil et al. [1] demonstrate that a network can be efficiently reconstructed with only a small subset of its original parameters.

Generally, there are two main branches of network pruning. One of them is unstructured pruning, also called weight pruning, which removes individual network connections. Han et al. [2] achieved a compression rate of 90% by pruning weights with small magnitudes and retraining the model. However, unstructured pruning produces sparse weight matrices, which cannot lead to actual speedup and compression without specialized hardware or libraries [3]. On the other hand, structured pruning methods eliminate the whole neuron or even the layer of the model, not individual connections. Since structured pruning maintains the original weight structure, no specialized hardware or libraries are necessary for acceleration. The most prevalent structured pruning method for CNN models is to prune filters of each convolution layer and the corresponding output feature map channels. The filter or channel to be removed is determined by various saliency criteria [15, 26, 27].

Regardless of what saliency criterion is used, the corresponding dimension of the pruned neuron is removed from the next layer. Consequently, the output of the next layer will not be fully reconstructed

---

∗Corresponding author

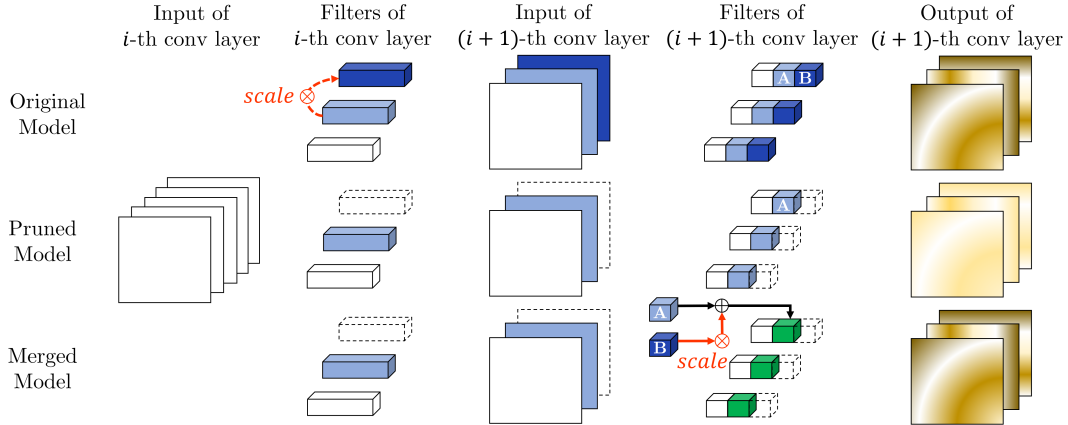

Figure 1: Neuron merging applied to the convolution layer. The pruned filter is marked as a dashed box. Pruning the blue-colored filter results in the removal of its corresponding feature map and the corresponding dimensions in the next layer, which leads to the scale-down of the output feature maps. However, neuron merging maintains the scale of the output feature maps by merging the pruned dimension (**B**) with the remaining one (**A**). Let us assume that the blue-colored filter is $scale$ times the light-blue-colored filter. By multiplying **B** by $scale$ and adding it to **A**, we can perfectly approximate the output feature map even after removing the blue-colored filter.

with the remaining neurons. In particular, when the neurons of the front layer are removed, the reconstruction error continues to accumulate, which leads to performance degradation [27].

In this paper, we propose neuron merging that compensates for the effect of the removed neuron by merging its corresponding dimension of the next layer. Neuron merging is applicable to both the fully connected and convolution layers, and the overall concept applied to the convolution layer is depicted in Fig. 1. Neuron merging starts with decomposing the original weights into two matrices/tensors. One of them becomes the new weights for the current layer, and the other is what we name a scaling matrix, guiding the process of merging the dimensions of the next layer. If the activation function is ReLU and the scaling matrix satisfies certain conditions, it can be absorbed into the next layer; thus, merging has the same network topology as pruning.

In this formulation, we also propose a simple and data-free method of neuron merging. To form the remaining weights, we utilize well-known pruning criteria (e.g., $l_1$-norm [15]). To generate the scaling matrix, we employ the cosine similarity and $l_2$-norm ratio between neurons. This method is applicable even when only the pretrained model is given without any training data. Our extensive experiments demonstrate the effectiveness of our approach. For VGG-16 [21] and WideResNet 40-4 [28] on CIFAR-10, we achieve an accuracy of 93.16% and 93.3% without any fine-tuning, while reducing 64% and 40% of the total parameters, respectively. Our contributions are as follows:

(1) We propose and formulate a novel concept of neuron merging that compensates for the information loss due to the pruned neurons/filters in both fully connected layers and convolution layers.

(2) We propose a one-shot and data-free method of neuron merging which employs the cosine similarity and ratio between neurons.

(3) We show that our merged model better preserves the original model than the pruned model with various measures, such as the accuracy immediately after pruning, feature map visualization, and Weighted Average Reconstruction Error [27].

## 2 Related Works

A variety of criteria [5, 6, 15, 18, 26, 27] have been proposed to evaluate the importance of a neuron, in the case of CNN, a filter. However, all of them suffer from significant accuracy drop immediately after the pruning. Therefore, fine-tuning the pruned model often requires as many epochs as training the original model to restore the accuracy near the original model. Several works [16, 25] add trainable parameters to each feature map channel to obtain data-driven channel sparsity, enabling

the model to automatically identify redundant filters. In this case, training the model from scratch is inevitable to obtain the channel sparsity, which is a time- and resource-consuming process.

Among filter pruning works, Luo et al. [17] and He et al. [7] have similar motivation to ours, aiming to similarly reconstruct the output feature map of the next layer. Luo et al. [17] search the subset of filters that have the smallest effect on the output feature map of the next layer. He et al. [7] propose LASSO regression based channel selection and least square reconstruction of output feature maps. In both papers, data samples are required to obtain feature maps. However, our method is novel in that it compensates for the loss of removed filters in a one-shot and data-free way.

Srinivas and Babu [22] introduce data-free neuron pruning for the fully connected layers by iteratively summing up the co-efficients of two similar neurons. Different from [22], neuron merging introduces a different formulation including the scaling matrix to systematically incorporate the ratio of neurons and is applicable to various model structures such as the convolution layer with batch normalization. More recently, Mussay et al. [19] approximate the output of the next layer by finding the coresets of neurons and discarding the rest.

"Pruning-at-initialization" methods [14, 23] prune individual connections in advance to save resources at training time. SNIP [14] and GraSP [23] use gradients to measure the importance of connections. In contrast, our approach is applied to structured pruning, so no specialized hardware or libraries are necessary to handle sparse connections. Also, our approach can be adopted even when the model is trained without any consideration of pruning.

Canonical Polyadic (CP) decomposition [12] and Tucker decomposition [10] are widely used to lighten convolution kernel tensor. At first glance, our method is similar to row rank approximation in that it starts with decomposing the weight matrix/tensor into two parts. Different from row rank approximation works, we do not retain all decomposed matrices/tensors during inference time. Instead, we combine one of the decomposed matrices with the next layer and achieve the same acceleration as structured network pruning.

## 3  Methodology

First, we mathematically formulate the new concept of neuron merging in the fully connected layer. Then, we show how merging is applied to the convolution layer. In Section 3.3, we introduce one possible data-free method of merging.

### 3.1  Fully Connected Layer

For simplicity, we start with the fully connected layer without bias. Let $N_i$ denote the length of input column vector for the $i$-th fully connected layer. The $i$-th fully connected layer transforms the input vector $\mathbf{x}_i \in \mathbb{R}^{N_i}$ into the output vector $\mathbf{x}_{i+1} \in \mathbb{R}^{N_{i+1}}$. The network weights of the $i$-th layer are denoted as $\mathbf{W}_i \in \mathbb{R}^{N_i \times N_{i+1}}$.

Our goal is to maintain the activation vector of the $(i + 1)$-th layer, which is

$$\mathbf{a}_{i+1} = \mathbf{W}_{i+1}^\top f \left( \mathbf{W}_i^\top \mathbf{x}_i \right), \tag{1}$$

where $f$ is an activation function.

Now, we decompose the weight matrix $\mathbf{W}_i$ into two matrices, $\mathbf{Y}_i \in \mathbb{R}^{N_i \times P_{i+1}}$ and $\mathbf{Z}_i \in \mathbb{R}^{P_{i+1} \times N_{i+1}}$, where $0 < P_{i+1} \leq N_{i+1}$. Therefore, $\mathbf{W}_i \approx \mathbf{Y}_i \mathbf{Z}_i$. Then Eq. 1 is approximated as,

$$\mathbf{a}_{i+1} \approx \mathbf{W}_{i+1}^\top f \left( \mathbf{Z}_i^\top \mathbf{Y}_i^\top \mathbf{x}_i \right). \tag{2}$$

The key idea of neuron merging is to combine $\mathbf{Z}_i$ and $\mathbf{W}_{i+1}$, the weight of the next layer. In order for $\mathbf{Z}_i$ to be moved out of the activation function, $f$ should be ReLU and a certain constraint on $\mathbf{Z}_i$ is necessary.

**Theorem 1.** *Let* $\mathbf{Z} \in \mathbb{R}^{P \times N}$, $\mathbf{v} \in \mathbb{R}^P$. *Then,*

$$f(\mathbf{Z}^\top \mathbf{v}) = \mathbf{Z}^\top f(\mathbf{v}), \quad \text{for all } \mathbf{v} \in \mathbb{R}^P,$$

*if and only if* $\mathbf{Z}$ *has only non-negative entries with at most one strictly positive entry per column.*

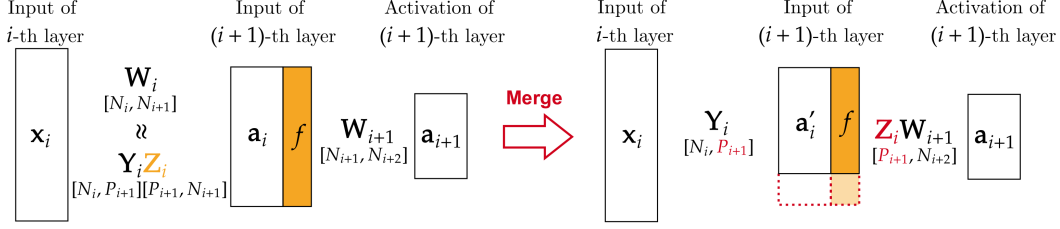

Figure 2: Illustration of the two consecutive fully connected layers before and after the merging step. After the merging, the number of neurons in the $(i+1)$-th layer decreases from $N_{i+1}$ to $P_{i+1}$ as matrix $Z_i$ is combined with the weight of the next layer.

See the Appendix for proof of Theorem 1. If $f$ is ReLU and $\mathbf{Z}_i$ satisfies the condition of Theorem 1, Eq. 2 is derived as,

$$\mathbf{a}_{i+1} \approx \mathbf{W}_{i+1}^\top \mathbf{Z}_i^\top f(\mathbf{Y}_i^\top \mathbf{x}_i) = (\mathbf{Z}_i \mathbf{W}_{i+1})^\top f(\mathbf{Y}_i^\top \mathbf{x}_i) = (\mathbf{W}_{i+1}')^\top f(\mathbf{Y}_i^\top \mathbf{x}_i), \qquad (3)$$

where $\mathbf{W}_{i+1}' = \mathbf{Z}_i \mathbf{W}_{i+1} \in \mathbb{R}^{P_{i+1} \times N_{i+2}}$. As shown in Fig. 2, the number of neurons in the $(i+1)$-th layer is reduced from $N_{i+1}$ to $P_{i+1}$ after merging, so the network topology is identical to that of structured pruning. Therefore, $\mathbf{Y}_i$ represents the new weights remaining in the $i$-th layer, and $\mathbf{Z}_i$ is the scaling matrix, indicating how to compensate for the removed neurons. We provide the same derivation for the fully connected layer with bias in the Appendix.

## 3.2 Convolution Layer

For merging in the convolution layer, we first define two operators for $N$-way tensors.

**n-mode product** According to Kolda and Bader [11], the *n-mode (matrix) product* of a tensor $\mathcal{X} \in \mathbb{R}^{I_1 \times I_2 \times \cdots \times I_N}$ with a matrix $\mathbf{U} \in \mathbb{R}^{J \times I_n}$ is denoted by $\mathcal{X} \times_n \mathbf{U}$ and is size of $I_1 \times \cdots \times I_{n-1} \times J \times I_{n+1} \times \cdots \times I_N$.

Elementwise, we have

$$[\mathcal{X} \times_n \mathbf{U}]_{i_1 \cdots i_{n-1} j i_{n+1} \cdots i_N} = \sum_{i_n=1}^{I_n} x_{i_1 i_2 \cdots i_N} u_{j i_n}.$$

**Tensor-wise convolution** We define tensor-wise convolution operator $\circledast$ between a 4-way tensor $\mathcal{W} \in \mathbb{R}^{N \times C \times K \times K}$ and a 3-way tensor $\mathcal{X} \in \mathbb{R}^{C \times H \times W}$. For simple notation, we assume that the stride of convolution is 1. However, this notation can be generalized to other convolution settings.

Elementwise, we have

$$[\mathcal{W} \circledast \mathcal{X}]_{\alpha \beta \gamma} = \sum_{c=1}^{C} \sum_{h=1}^{K} \sum_{w=1}^{K} \mathcal{W}_{\alpha c h w} \mathcal{X}_{c(h+\beta-1)(w+\gamma-1)}.$$

Intuitively, $\mathcal{W} \circledast \mathcal{X}$ denotes the channel-wise concatenation of the output feature map matrices that result from 3D convolution operation between $\mathcal{X}$ and each filter of $\mathcal{W}$.

**Merging the convolution layer** Now we extend neuron merging to the convolution layer. Similar to the fully connected layer, let $N_i$ and $N_{i+1}$ denote the number of input and output channels of the $i$-th convolution layer. The $i$-th convolution layer transforms the input feature map $\mathcal{X}_i \in \mathbb{R}^{N_i \times H_i \times W_i}$ into the output feature map $\mathcal{X}_{i+1} \in \mathbb{R}^{N_{i+1} \times H_{i+1} \times W_{i+1}}$. The filter weights of the $i$-th layer are denoted as $\mathcal{W}_i \in \mathbb{R}^{N_{i+1} \times N_i \times K \times K}$ which consists of $N_{i+1}$ filters.

Our goal is to maintain the activation feature map of the $(i+1)$-th layer, which is

$$\mathcal{A}_{i+1} = \mathcal{W}_{i+1} \circledast f\left(\mathcal{W}_i \circledast \mathcal{X}_i\right). \qquad (4)$$

We decompose the 4-way tensor $\mathcal{W}_i$ into a matrix $\mathbf{Z}_i \in \mathbb{R}^{P_{i+1} \times N_{i+1}}$ and a 4-way tensor $\mathcal{Y}_i \in \mathbb{R}^{P_{i+1} \times N_i \times K \times K}$. Therefore,

$$\mathcal{W}_i \approx \mathcal{Y}_i \times_1 \mathbf{Z}_i^\top. \qquad (5)$$

Then Eq. 4 is approximated as,

$$\mathcal{A}_{i+1} \approx \mathcal{W}_{i+1} \circledast f\left(\left(\mathcal{Y}_i \times_1 \mathbf{Z}_i^{\top}\right) \circledast \mathcal{X}_i\right)$$
$$= \mathcal{W}_{i+1} \circledast f\left(\left(\mathcal{Y}_i \circledast \mathcal{X}_i\right) \times_1 \mathbf{Z}_i^{\top}\right). \tag{6a}$$

The key idea of neuron merging is to combine $\mathbf{Z}_i$ and $\mathcal{W}_{i+1}$, the weight of the next layer. If $f$ is ReLU, we can extend Theorem 1 to a 1-mode product of tensor.

**Corollary 1.1.** *Let* $\mathbf{Z} \in \mathbb{R}^{P \times N}$, $\mathcal{X} \in \mathbb{R}^{P \times H \times W}$. *Then,*

$$f(\mathcal{X} \times_1 \mathbf{Z}^{\top}) = f(\mathcal{X}) \times_1 \mathbf{Z}^{\top}, \quad \text{for all } \mathcal{X} \in \mathbb{R}^{P \times H \times W},$$

*if and only if* $\mathbf{Z}$ *has only non-negative entries with at most one strictly positive entry per column.*

If $f$ is ReLU and $\mathbf{Z}_i$ satisfies the condition of Corollary 1.1,

$$\mathcal{A}_{i+1} \approx \mathcal{W}_{i+1} \circledast \left(f\left(\mathcal{Y}_i \circledast \mathcal{X}_i\right) \times_1 \mathbf{Z}_i^{\top}\right)$$
$$= \left(\mathcal{W}_{i+1} \times_2 \mathbf{Z}_i\right) \circledast f\left(\mathcal{Y}_i \circledast \mathcal{X}_i\right) \tag{7a}$$
$$= \mathcal{W}_{i+1}' \circledast f\left(\mathcal{Y}_i \circledast \mathcal{X}_i\right),$$

where $\mathcal{W}_{i+1}' = (\mathcal{W}_{i+1} \times_2 \mathbf{Z}_i) \in \mathbb{R}^{N_{i+2} \times P_{i+1} \times K \times K}$. See the Appendix for proofs of Corollary 1.1, Eq. 6a, and 7a. After merging, the number of filters in the $i$-th convolution layer is reduced from $N_{i+1}$ to $P_{i+1}$, so the network topology is identical to that of structured pruning. As $\mathbf{Z}_i$ is merged with the weights of the $(i+1)$-th layer, the pruned dimensions are absorbed into the remaining ones, as shown in Fig 1.

### 3.3 Proposed Algorithm

The overall process of neuron merging is as follows. First, we decompose the weights into two parts. $\mathbf{Y}_i/\mathcal{Y}_i$ represents the new weights remaining in the $i$-th layer, and $\mathbf{Z}_i$ is the scaling matrix. After the decomposition, $\mathbf{Z}_i$ is combined with the weights of the next layer, as described in Section 3.1 and 3.2. Therefore, the actual compensation takes place by merging the dimensions of the next layer. The corresponding dimension of a pruned neuron is multiplied by a positive number and then added to that of the retained neuron.

Now we propose a simple one-shot method to decompose the weight matrix/tensor into two parts. First, we select the most useful neurons to form $\mathbf{Y}_i/\mathcal{Y}_i$. We can utilize any pruning criteria. Then, we generate $\mathbf{Z}_i$ by selecting the most similar remaining neuron for each pruned neuron and measuring the ratio between them. Algorithm 1 describes the overall procedure of decomposition for the case of one-dimensional neurons in a fully connected layer. The same algorithm is applied to the convolution filters after reshaping each three-dimensional filter tensor to a one-dimensional vector.

According to Theorem 1, if a pruned neuron can be expressed as a positive multiple of a remaining one, we can remove and compensate for it without causing any loss in the output vector. This gives us an important insight into the criterion for determining similar neurons: direction, not absolute distance. Therefore, we employ the cosine similarity to select similar neurons. Algorithm 2 demonstrates selecting the most similar neuron with the given one and obtaining the scale between them. We set the scale value as an $l_2$-norm ratio of the two neurons. The scale value indicates how much to compensate for the removed neuron in the following layer.

Here we introduce a hyperparameter $t$; we compensate only when the similarity between the two neurons is above $t$. If $t$ is -1, all pruned neurons are compensated for, and the number of compensated neurons decreases as $t$ approaches 1. If none of the removed neurons is compensated for, the result is exactly the same as vanilla pruning. In other words, pruning can be considered as a special case of neuron merging.

**Batch normalization layer** For modern CNN architectures, batch normalization [9] is widely used to prevent an internal covariate shift. If batch normalization is applied after a convolution layer, the output feature map channels of two identical filters could be different. Therefore, we introduce an additional term to consider when selecting the most similar filter.

Let $\mathcal{X} \in \mathbb{R}^{c \times h \times w}$ denote the output feature map of a convolution layer, and $\mathcal{X}^{BN} \in \mathbb{R}^{c \times h \times w}$ denote $\mathcal{X}$ after a batch normalization layer. The batch normalization layer contains four types of parameters, $\boldsymbol{\gamma}, \boldsymbol{\beta}, \boldsymbol{\mu}, \boldsymbol{\sigma} \in \mathbb{R}^c$.

For simplicity, we consider the element-wise scale of two feature maps. Let $x_1^{BN} = \mathcal{X}_{1,1,1}^{BN}$, $x_2^{BN} = \mathcal{X}_{2,1,1}^{BN}$, $x_1 = \mathcal{X}_{1,1,1}$, $x_2 = \mathcal{X}_{2,1,1}$. Let $s$ denote the $l_2$-norm ratio of $\mathcal{X}_{1,:,:}$ and $\mathcal{X}_{2,:,:}$. Assuming that they have the same direction, the relationship between $x_1^{BN}$ and $x_2^{BN}$ is as follows:

$$x_1^{BN} = \gamma_1\left(\frac{x_1 - \mu_1}{\sigma_1}\right) + \beta_1, \quad x_2^{BN} = \gamma_2\left(\frac{x_2 - \mu_2}{\sigma_2}\right) + \beta_2, \quad x_2 = s \times x_1.$$

$$x_2^{BN} = \mathcal{S} \times x_1^{BN} + \mathcal{B}, \quad \text{where } \mathcal{S} := s\frac{\gamma_2}{\gamma_1}\frac{\sigma_1}{\sigma_2}, \quad \mathcal{B} := \frac{\gamma_2}{\sigma_2}\left[s\left(-\frac{\sigma_1\beta_1}{\gamma_1} + \mu_1\right) - \mu_2\right] + \beta_2. \quad (8)$$

According to Eq. 8, if $\mathcal{B}$ is 0, the ratio of $x_2^{BN}$ to $x_1^{BN}$ is exactly $\mathcal{S}$. Therefore, we select the filter that simultaneously minimizes the cosine distance ($1 - CosineSim$) and the bias distance ($|\mathcal{B}|/\mathcal{S}$) and then use $\mathcal{S}$ as $scale$. We normalize the bias distance between 0 and 1. The overall selection procedure for a convolution layer with batch normalization is described in Algorithm 3. The input includes the $n$-th filter of the convolution layer, denoted as $\mathcal{F}_n$. A hyperparameter $\lambda$ is employed to control the ratio between the cosine distance and the bias distance.

---

**Algorithm 1** Decomposition Algorithm

**Input:** $\mathbf{W}_i \in \mathbb{R}^{N_i \times N_{i+1}}$
**Given:** $P_{i+1}, t$
   $\mathbf{Y}_i \leftarrow$ set of $P_{i+1}$ selected neurons
   Initialize $\mathbf{Z}_i \in \mathbb{R}^{P_{i+1} \times N_{i+1}}$ with 0
   **for** every neuron $\mathbf{w}_n \in \mathbb{R}^{N_i}$ in $\mathbf{W}_i$ **do**
      **if** $\mathbf{w}_n \in \mathbf{Y}_i$ **then**
         $p \leftarrow$ index of $\mathbf{w}_n$ within $\mathbf{Y}_i$
         $z_{pn} \leftarrow 1$
      **else**
         $\mathbf{w}_n^*, sim, scale \leftarrow MostSim(\mathbf{w}_n, \mathbf{Y}_i)$
         $p^* \leftarrow$ index of $\mathbf{w}_n^*$ within $\mathbf{Y}_i$
         **if** $sim \geq t$ **then**
            $z_{p^*n} \leftarrow scale$
         **end if**
      **end if**
   **end for**
**Output:** $\mathbf{Y}_i \in \mathbb{R}^{N_i \times P_{i+1}}, \mathbf{Z}_i \in \mathbb{R}^{P_{i+1} \times N_{i+1}}$

---

**Algorithm 2** MostSim Algorithm

**Input:** $\mathbf{w}_n \in \mathbb{R}^{N_i}, \mathbf{Y}_i \in \mathbb{R}^{N_i \times P_{i+1}}$
   $\mathbf{w}_n^* \leftarrow \arg\max_{\mathbf{w} \in \mathbf{Y}_i} CosineSim(\mathbf{w}_n, \mathbf{w})$
   $sim \leftarrow CosineSim(\mathbf{w}_n, \mathbf{w}_n^*)$
   $scale \leftarrow \|\mathbf{w}_n\|_2 / \|\mathbf{w}_n^*\|_2$
**Output:** $\mathbf{w}_n^* \in \mathbb{R}^{N_i}, sim, scale$

---

**Algorithm 3** MostSim Algorithm with BN

**Input:** $\mathcal{F}_n \in \mathbb{R}^{N_i \times K \times K}, \mathcal{Y}_i \in \mathbb{R}^{P_{i+1} \times N_i \times K \times K}$,
   $\boldsymbol{\mu}_i, \boldsymbol{\sigma}_i, \boldsymbol{\gamma}_i, \boldsymbol{\beta}_i \in \mathbb{R}^{N_{i+1}}$
**Given:** $\lambda$
   **for** $m$ in $[1, P_{i+1}]$ **do**
      $CosList[m] \leftarrow 1 - CosineSim(\mathcal{F}_n, \mathcal{F}_m)$
      $BiasList[m] \leftarrow (|\mathcal{B}|/\mathcal{S})$ in Eq. 8
   **end for**
   $Normalize(BiasList)$
   $DistList \leftarrow CosList \times \lambda + BiasList \times (1 - \lambda)$
   $\mathcal{F}_n^* \leftarrow \arg\min_{\mathcal{F}_m \in \mathcal{Y}_i} DistList[m]$
   $sim \leftarrow CosineSim(\mathcal{F}_n, \mathcal{F}_n^*)$
   $scale \leftarrow \mathcal{S}$ in Eq. 8
**Output:** $\mathcal{F}_n^* \in \mathbb{R}^{N_i \times K \times K}, sim, scale$

---

## 4 Experiments

Neuron merging aims to preserve the original model by maintaining the scale of the output feature map better than network pruning. To validate this, we compare the initial accuracy, feature map visualization, and Weighted Average Reconstruction Error [27] of image classification, without fine-tuning. We evaluate the proposed approach with several popular models, which are LeNet [13], VGG [21], ResNet [4], and WideResNet [28], on FashionMNIST [24], CIFAR [8], and ImageNet[1] [20] datasets. We use FashionMNIST instead of MNIST [13] as the latter classification is rather simple compared to the capacity of the LeNet-300-100 model. As a result, it is difficult to check the accuracy degradation after pruning with MNIST.

To train the baseline models, we employ SGD with the momentum of 0.9. The learning rate starts at 0.1, with different annealing strategies per model. For LeNet, the learning rate is reduced by one-tenth for every 15 of the total 60 epochs. Weight decay is set to 1e-4, and batch size to 128. For VGG and ResNet, the learning rate is reduced by one-tenth at 100 and 150 of the total 200 epochs. Weight decay is set to 5e-4, and batch size to 128. Weights are randomly initialized before the training. To preprocess FashionMNIST images, each one is normalized with a mean and standard deviation of 0.5; for CIFAR, we follow the setting in He et al. [6].

Table 1: Performance comparison of pruning and merging for LeNet-300-100 on FashionMNIST without fine-tuning. 'Acc.↑' denotes the accuracy gain of merging compared to pruning.

| Pruning Ratio | Baseline Acc. | $l_1$-norm | | | $l_2$-norm | | | $l_2$-GM | | |
|---|---|---|---|---|---|---|---|---|---|---|
| | | Prune | Merge | Acc. ↑ | Prune | Merge | Acc. ↑ | Prune | Merge | Acc. ↑ |
| 50% | | 88.40% | **88.69%** | 0.29% | 87.86% | **88.38%** | 0.52% | 88.08% | **88.57%** | 0.49% |
| 60% | 89.80% | 85.17% | **86.92%** | 1.75% | 83.03% | **88.07%** | 5.04% | 85.82% | **88.10%** | 2.28% |
| 70% | | 71.26% | **82.75%** | 11.49% | 71.21% | **83.27%** | 12.06% | 78.38% | **86.39%** | 8.01% |
| 80% | | 66.76% | **80.02%** | 13.26% | 63.90% | **77.11%** | 13.21% | 64.19% | **77.49%** | 13.30% |

Table 2: Performance comparison of pruning and merging for VGG-16 on CIFAR datasets without fine-tuning. '**M-P**' denotes the accuracy recovery of merging compared to pruning. '**B-M**' denotes the accuracy drop of the merged model compared to the baseline model. 'Param. ↓ (#)' denotes the parameter reduction rate and the absolute number of pruned/merged models.

| Dataset | Criterion | Baseline Acc. (**B**) | Initial Acc. | | | **B-M** | Param. ↓ (#) |
|---|---|---|---|---|---|---|---|
| | | | Prune(**P**) | Merge(**M**) | **M-P** | | |
| CIFAR-10 | $l_1$-norm | | 88.70% | **93.16%** | 4.46% | 0.54% | |
| | $l_2$-norm | 93.70% | 89.14% | **93.16%** | 4.02% | 0.54% | 63.7% (5.4M) |
| | $l_2$-GM | | 87.85% | **93.10%** | 5.25% | 0.60% | |
| CIFAR-100 | $l_1$-norm | | 67.70% | **71.63%** | 3.93% | 1.67% | |
| | $l_2$-norm | 73.30% | 67.79% | **71.86%** | 4.07% | 1.44% | 44.1% (8.4M) |
| | $l_2$-GM | | 67.38% | **71.95%** | 4.57% | 1.35% | |

In Section 3.3, we introduced two hyperparameters for neuron merging: $t$ and $\lambda$. For $t$, we use 0.45 for LeNet, and 0.1 for other convolution models. For $\lambda$, the value between 0.7 and 0.9 generally gives a stable performance. Specifically, we use 0.85 for VGG and ResNet on CIFAR-10, 0.8 for WideResNet on CIFAR-10, and 0.7 for VGG-16 on CIFAR-100.

We test neuron merging with three structured pruning criteria: 1) '$l_1$-norm' proposed in [15]; 2) '$l_2$-norm' proposed in [5]; and 3) '$l_2$-GM' proposed in [6], referring to pruning filters with a small $l_2$ distance from the geometric median. These methods were originally proposed for convolution filters but can be applied to the neurons in fully connected layers. Among various pruning criteria, these methods have the top-level initial accuracy. In accordance with the data-free characteristic of our method, we exclude pruning methods that require feature maps or data loss in filter scoring.

## 4.1 Initial Accuracy of Image Classification

**LeNet-300-100** The results of LeNet-300-100 with bias on FashionMNIST are presented in Table 1. The number of neurons in each layer is reduced in proportion to the pruning ratio. As shown in Table 1, the pruned model's performance deteriorates as more neurons are pruned. However, if the removed neurons are compensated for with merging, the performance improves in all cases. Accuracy gain is more prominent as the pruning ratio increases. For example, when the pruning ratio is 80%, the merging recovers more than 13% of accuracy compared to the pruning.

**VGG-16** We test neuron merging for VGG-16 on CIFAR datasets. As described in Table 2, the merging shows an impressive accuracy recovery on both datasets. For CIFAR-10, we adopt the pruning strategy from PFEC [15], pruning half of the filters in the first convolution layer and the last six convolution layers. Compared to the baseline model, the accuracy after pruning is dropped by 5% on average with a parameter reduction of 63%. On the other hand, merging improves the accuracy to a near-baseline level for all three pruning criteria, showing a mere 0.6% drop at most.

For CIFAR-100, we slightly modified the pruning strategy of PFEC. In addition to the first convolution layer, we prune only the last three, not six, convolution layers. With this strategy, we can still reduce 44.1% of total parameters. Similar to CIFAR-10, the merging recovers about 4% of the performance deterioration caused by the pruning. In CIFAR-100, the accuracy drop compared to the baseline was about 1% greater than CIFAR-10. This seems to be because the filter redundancy decreases as the target label diversifies. Interestingly, the accuracy gain of merging is most prominent in the '$l_2$-GM' [6] criterion, and the final accuracy is also the highest.

**ResNet** We also test our neuron merging for ResNet-56 and WideResNet-40-4, on CIFAR-10. We additionally adopt WideResNet-40-4 to examine the effect of merging with extra channel redundancy.

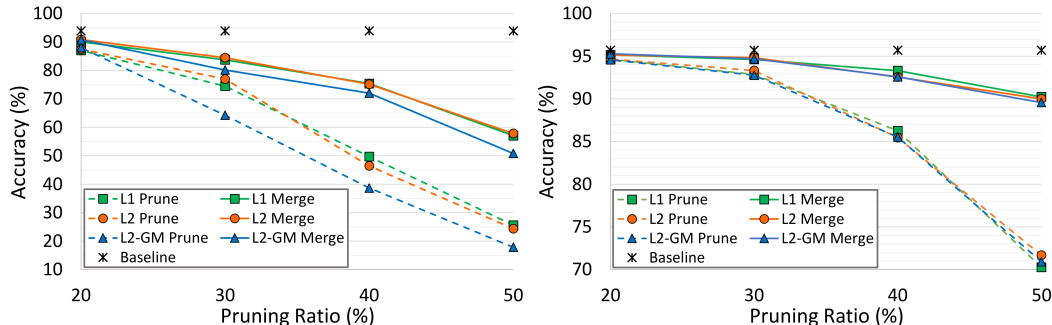

Figure 3: Performance analysis of ResNet-56 (left) and WideResNet-40-4 (right) on CIFAR-10 under different pruning ratios. Dashed lines indicate the accuracy trend of pruning, and solid lines indicate that of merging. Black asterisks indicate the accuracy of the baseline model.

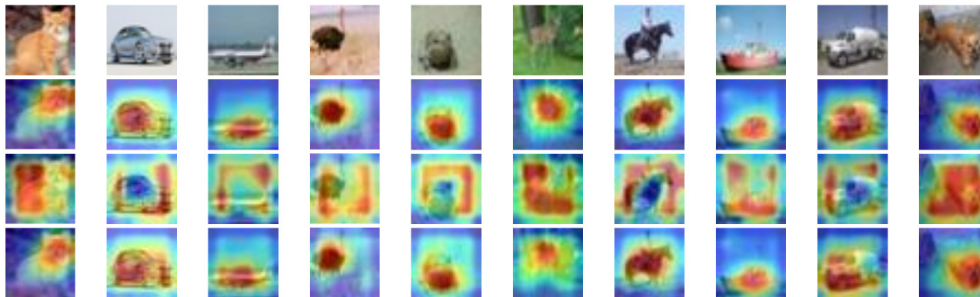

Figure 4: Feature map visualization of WideResNet-40-4. The top row is the original image, and the feature maps of the baseline model, pruned model, and merged model are in order. We select one image for each image label.

To avoid the misalignment of feature map in the shortcut connection, we only prune the internal layers of the residual blocks as in [15, 17]. We carry out experiments on four different pruning ratios: 20%, 30%, 40%, and 50%. The pruning ratio refers to how many filters are pruned in each internal convolution layer.

As shown in Fig. 3, ResNet-56 noticeably suffer from performance deterioration in all pruning cases because of its narrow structure. However, the merging increases the accuracy in all cases. As the pruning ratio increases, merging exhibits a more prominent recovery. When the pruning ratio is 50%, merging restores accuracy by more than 30%. Since, structurally, ResNet has insufficient channels to reuse, the merging alone has limits in recovery. After fine-tuning, both the pruned and merged models reach comparable accuracy. Interestingly, the '$l_2$-GM' criterion shows a more significant accuracy drop than other norm-based criteria after pruning and merging. On the other hand, for WideResNet, three pruning criteria show a similar trend in accuracy drop. As the pruning ratio increases, the accuracy trend of merging falls more gradually than pruning. Since the number of compensable channels increases in WideResNet, the accuracy after merging is closer to baseline accuracy than ResNet. Even after removing 50% of the filters, the merging only shows an accuracy loss of less than 5%, which is 20% better than pruning.

## 4.2 Feature Map Reconstruction of Neuron Merging

To further validate that merging better preserves the original feature maps than pruning, we make use of two types of measures, namely feature map visualization and Weighted Average Reconstruction Error. We visualize the output feature map of the last residual block in WideResNet-40-4 on CIFAR-10. Fifty percent of the total filters are pruned with $l_1$-norm criterion. Feature maps are resized in the same way as [29]. As shown in Fig. 4, while the original model captures the coarse-grained area of the object, the pruned model produces noisy and divergent feature maps. However, the feature maps of our merged model are very similar to those of the original model. Although the heated regions are slightly blurrier than in the original model, the merged model accurately detects the object area.

Table 3: WARE comparison of pruning and merging for various models on CIFAR-10. 'WARE ↓' denotes the WARE drop of the merged model compared to the pruned model.

| Model | Pruning Ratio | $l_1$-norm | | | $l_2$-norm | | | $l_2$-GM | | |
|---|---|---|---|---|---|---|---|---|---|---|
| | | Prune | Merge | WARE ↓ | Prune | Merge | WARE ↓ | Prune | Merge | WARE ↓ |
| VGG-16 | - | 4.285 | **1.465** | 2.820 | 4.394 | **1.555** | 2.839 | 4.515 | **1.599** | 2.916 |
| ResNet-56 | 50% | 12.095 | **4.986** | 7.109 | 12.566 | **4.352** | 8.214 | 11.691 | **5.679** | 6.012 |
| | 40% | 8.759 | **3.911** | 4.848 | 9.094 | **3.416** | 5.678 | 10.099 | **4.264** | 5.835 |
| | 30% | 6.251 | **3.646** | 2.605 | 5.224 | **3.556** | 1.668 | 6.888 | **3.568** | 3.320 |
| | 20% | 3.748 | **2.508** | 1.240 | 3.745 | **2.382** | 1.363 | 3.685 | **2.448** | 1.237 |
| WideResNet-40-4 | 50% | 3.502 | **2.364** | 1.138 | 3.446 | **2.406** | 1.040 | 3.515 | **2.498** | 1.017 |
| | 40% | 2.849 | **1.649** | 1.200 | 2.921 | **1.821** | 1.100 | 2.868 | **1.714** | 1.154 |
| | 30% | 2.099 | **1.213** | 0.886 | 2.129 | **1.315** | 0.814 | 2.168 | **1.271** | 0.897 |
| | 20% | 1.266 | **0.796** | 0.470 | 1.103 | **0.754** | 0.349 | 1.161 | **0.746** | 0.415 |

Weighted Average Reconstruction Error (WARE) is proposed in [27] to measure the change of the important neurons' responses on the final response layer after pruning (without fine-tuning). The final response layer refers to the second-to-last layer before classification. WARE is defined as

$$\text{WARE} = \frac{\sum_{m=1}^{M} \sum_{i=1}^{N} s_i \cdot \frac{|\hat{y}_{i,m} - y_{i,m}|}{|y_{i,m}|}}{M \cdot N}, \tag{9}$$

where $M$ and $N$ represent the number of samples and number of retained neurons in the final response layer, respectively; $s_i$ is the importance score of the $i$-th neuron; and $y_{i,m}$ and $\hat{y}_{i,m}$ are the responses on the $m$-th sample of the $i$-th neuron before/after pruning.

Neuron importance scores ($s_i$) are set to 1 to reflect the effect of all neurons equally. Therefore, the lower the WARE is, the more the network output (i.e., logit values) is similar to that of the original. We measure the WARE of all three kinds of models presented in Section 4.1 on CIFAR-10. Our merged model has lower WARE than the pruned model in all cases. Similar with the initial accuracy, the WARE drops considerably as the pruning ratio increases. We provide a detailed result in Table 3. Through these experiments, we can validate that neuron merging compensates well for the removed neurons and approximates the output feature map of the original model.

## 5 Conclusion

In this paper, we propose and formulate a novel concept of neuron merging that compensates for the accuracy loss of the pruned neurons. Our one-shot and data-free method better reconstructs the output feature maps of the original model than vanilla pruning. To demonstrate the effectiveness of merging over network pruning, we compare the initial accuracy, WARE, and feature map visualization on image-classification tasks. It is worth noting that decomposing the weights can be varied in the neuron merging formulation. We will explore the possibility of improving the decomposition algorithm. Furthermore, we plan to generalize the neuron merging formulation to more diverse activation functions and model architectures.

## Broader Impact

This work has the same potential impact as any neural network acceleration study. The positive effect comes from reducing the resource overhead of deep learning models during inference time. Data-free acceleration approaches have more potential in that the model can be lightened using only model weights, without any access to the training dataset. Therefore, we can more easily deploy neural network models to mobile phones or edge devices. We thus take a step closer to energy-friendly deep learning, facilitating a wider use of Artificial Intelligence in industrial IoT or Smart-home technology.

At the same time, research on neural network acceleration may have some negative consequences. If the neural network models are more widely used for wearable devices or surveillance cameras, there is a possibility of privacy invasion or cybercrime. In addition, the malfunction of industrial IoT devices could cause a severe problem for the whole production process.

## Acknowledgments and Disclosure of Funding

This research was results of a study on the "HPC Support" Project, supported by the 'Ministry of Science and ICT' and NIPA. This work was also supported by Korea Institute of Science and Technology (KIST) under the project "HERO Part 1: Development of core technology of ambient intelligence for proactive service in digital in-home care".

## Footnotes

[1]Test results on ImageNet are provided in the Appendix.

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
