[Supplementary Material]

# 6 APPENDIX

This appendix provides the mathematical proofs of the theoretical results and additional experiment results of our paper "Neuron Merging: Compensating for Pruned Neurons," accepted at 34th Conference on Neural Information Processing Systems (NeurIPS 2020).

## 6.1 Fully Connected Layer with Bias

The overall derivation is the same as Section 3.1. The difference is that we decompose the weights after concatenating the bias vector at the end of the weight matrix. Let $\mathbf{x}_i^B = [\mathbf{x}_i^\top | 1]^\top \in \mathbb{R}^{N_i+1}$, and $\mathbf{W}_i^B = [\mathbf{W}_i^\top | \mathbf{b}_i]^\top \in \mathbb{R}^{(N_i+1)\times N_{i+1}}$.

Our goal is to maintain the activation vector of the $(i+1)$-th layer, which is

$$\mathbf{a}_{i+1} = \mathbf{W}_{i+1}^\top f(\mathbf{W}_i^\top \mathbf{x}_i + \mathbf{b}_i) = \mathbf{W}_{i+1}^\top f((\mathbf{W}_i^B)^\top \mathbf{x}_i^B), \tag{10}$$

where $f$ is an activation function. Then, we decompose $\mathbf{W}_i^B$ into two matrices, $\mathbf{Y}_i^B \in \mathbb{R}^{(N_i+1)\times P_{i+1}}$, and $\mathbf{Z}_i^B \in \mathbb{R}^{P_{i+1}\times N_{i+1}}$, where $0 < P_{i+1} \leq N_{i+1}$. Therefore, $\mathbf{W}_i^B \approx \mathbf{Y}_i^B \mathbf{Z}_i^B$. Then Eq. 10 is approximated as,

$$\mathbf{a}_{i+1} \approx \mathbf{W}_{i+1}^\top f((\mathbf{Z}_i^B)^\top (\mathbf{Y}_i^B)^\top \mathbf{x}_i^B). \tag{11}$$

If $f$ is ReLU and $\mathbf{Z}_i$ satisfies the condition of Theorem 1,

$$\begin{aligned}
\mathbf{a}_{i+1} &\approx \mathbf{W}_{i+1}^\top (\mathbf{Z}_i^B)^\top f((\mathbf{Y}_i^B)^\top \mathbf{x}_i^B) \\
&= (\mathbf{Z}_i^B \mathbf{W}_{i+1})^\top f((\mathbf{Y}_i^{pruned})^\top \mathbf{x}_i + \mathbf{b}_i^{pruned}),
\end{aligned} \tag{12}$$

where $\mathbf{Y}_i^B = [(\mathbf{Y}_i^{pruned})^\top | \mathbf{b}_i^{pruned}]^\top$. $\mathbf{Y}_i^{pruned}$ and $\mathbf{b}_i^{pruned}$ denote the new weight and bias for the merged model, respectively. After merging, bias vector is detached from the weight matrix as the original model. Therefore, the number of neurons in the $(i+1)$-th layer is reduced from $N_{i+1}$ to $P_{i+1}$, and the corresponding entries of the bias vector are removed as well.

## 6.2 Proof of Theorem 1

**Theorem 1.** *Let* $\mathbf{Z} \in \mathbb{R}^{P\times N}$, $\mathbf{v} \in \mathbb{R}^P$. *Then,*

$$f(\mathbf{Z}^\top \mathbf{v}) = \mathbf{Z}^\top f(\mathbf{v}), \quad \text{for all } \mathbf{v} \in \mathbb{R}^P,$$

*if and only if* $\mathbf{Z}$ *has only non-negative entries with at most one strictly positive entry per column.*

*Proof.*

$$f(\mathbf{Z}^\top \mathbf{v}) = f(\mathbf{Z}^\top) f(\mathbf{v}) \tag{13a}$$
$$= \mathbf{Z}^\top f(\mathbf{v}). \tag{13b}$$

Let $z_{pn}$ denote the $(p, n)$ element of $\mathbf{Z}$, and $v_p$ denote the $p$-th element of $\mathbf{v}$. We also denote the $n$-th column vector of $\mathbf{Z}$ as $\mathbf{z}_n$. Eq. 13b is satisfied if and only if all the entries $z_{pn}$ are non-negative.

*Claim* 1.1. If $\mathbf{Z}$ has only non-negative entries with at most one strictly positive entry per column, then Eq. 13a also holds.

*Proof of Claim 1.1.* Let us define $I(n)$ as the row-index of the strictly positive entry in $\mathbf{z}_n$, or 1 if $\mathbf{z}_n = \mathbf{0}$.

$$\begin{aligned}
\mathbf{Z}^\top \mathbf{v} &= \left( z_{I(n)n} v_{I(n)} \right)^\top. \\
f(\mathbf{Z}^\top \mathbf{v}) &= \left( max(z_{I(n)n} v_{I(n)}, 0) \right)^\top \\
&= \left( z_{I(n)n} max(v_{I(n)}, 0) \right)^\top \\
&= \mathbf{Z}^\top f(\mathbf{v}).
\end{aligned}$$

We used the fact that $z_{I(n)n}$ is non-negative in the above equation.

*Claim* 1.2. If there exists a column with more than one strictly positive entry, then Eq. 13a does not hold in general.

*Proof of Claim 1.2.* Without loss of generality, say $\mathbf{z}_1$ has $K$ positive entries, $z_{11}, \cdots, z_{K1}$, where $2 \le K \le P$, and 0 otherwise. Also, we can assume that $z_{11} \ge z_{21} \ge \cdots \ge z_{K1} > 0$. Suppose one $\mathbf{v}$ which is,

$$
\mathbf{v} = (v_p)^\top = \begin{cases} -1, & 1 \le p < K \\ \frac{1}{2}, & p = K \\ 0, & K < p \le P. \end{cases}
$$

Then the first entry of $f(\mathbf{Z}^\top \mathbf{v})$ is equal to 0. However, the first entry of $f(\mathbf{Z}^\top) f(\mathbf{v})$ is equal to $z_{K1}/2$ which is not zero. Therefore, $f(\mathbf{Z}^\top \mathbf{v}) \ne f(\mathbf{Z}^\top) f(\mathbf{v})$.

∎

## 6.3 Proof of Corollary 1.1

**Corollary 1.1.** *Let* $\mathbf{Z} \in \mathbb{R}^{P \times N}$, $\mathcal{X} \in \mathbb{R}^{P \times H \times W}$. *Then,*

$$
f(\mathcal{X} \times_1 \mathbf{Z}^\top) = f(\mathcal{X}) \times_1 \mathbf{Z}^\top, \quad \text{for all } \mathcal{X} \in \mathbb{R}^{P \times H \times W},
$$

*if and only if* $\mathbf{Z}$ *has only non-negative entries with at most one strictly positive entry per column.*

*Proof.* According to Kolda and Bader [11], the definition of n-mode product is multiplying each mode-$n$ fiber of tensor $\mathcal{X}$ by the matrix $\mathbf{U}$. The idea can also be expressed in terms of unfolded tensors:

$$
\mathcal{Y} = \mathcal{X} \times_n \mathbf{U} \Leftrightarrow \mathbf{Y}_{(n)} = \mathbf{U} \mathbf{X}_{(n)}, \tag{14}
$$

where $X_{(n)}$ denotes mode-$n$ matricization of tensor $\mathcal{X}$.

Let $\mathcal{M} = f(\mathcal{X} \times_1 \mathbf{Z}^\top)$. If we re-express $\mathcal{M}$ referring to Eq. 14,

$$
\begin{aligned}
\mathbf{M}_{(1)} &= f(\mathbf{Z}^\top \mathbf{X}_{(1)}) \\
&= f(\mathbf{Z}^\top) f(\mathbf{X}_{(1)}) \\
&= \mathbf{Z}^\top f(\mathbf{X}_{(1)}).
\end{aligned}
$$

(15a) (15b)

Therefore, $\mathcal{M} = f(\mathcal{X}) \times_1 \mathbf{Z}^\top$. *Claim* 1.1 and *Claim* 1.2 can be generalized to prove that Eq. 15a and Eq. 15b holds.

∎

## 6.4 Proof of Equation 6a

**Equation 6a.** $(\mathcal{Y}_i \times_1 \mathbf{Z}_i^\top) \circledast \mathcal{X}_i = (\mathcal{Y}_i \circledast \mathcal{X}_i) \times_1 \mathbf{Z}_i^\top.$

*Proof.* For simple notation, subscript $i$ is omitted.

$$
\begin{aligned}
\left[ (\mathcal{Y} \times_1 \mathbf{Z}^\top) \circledast \mathcal{X} \right]_{\alpha\beta\gamma} &= \sum_{c=1}^{N} \sum_{h=1}^{K} \sum_{w=1}^{K} \left[ \mathcal{Y} \times_1 \mathbf{Z}^\top \right]_{\alpha chw} \mathcal{X}_{c(h+\beta-1)(w+\gamma-1)} \\
&= \sum_{c=1}^{N} \sum_{h=1}^{K} \sum_{w=1}^{K} \left( \sum_{\delta=1}^{P} \mathcal{Y}_{\delta chw} \mathbf{Z}_{\alpha\delta}^\top \right) \mathcal{X}_{c(h+\beta-1)(w+\gamma-1)} \\
&= \sum_{c=1}^{N} \sum_{h=1}^{K} \sum_{w=1}^{K} \sum_{\delta=1}^{P} \mathcal{Y}_{\delta chw} \mathbf{Z}_{\alpha\delta}^\top \mathcal{X}_{c(h+\beta-1)(w+\gamma-1)}.
\end{aligned} \tag{16}
$$

$$
\begin{aligned}
\left[ (\mathcal{Y} \circledast \mathcal{X}) \times_1 \mathbf{Z}^\top \right]_{\alpha\beta\gamma} &= \sum_{\delta=1}^{P} \left[ \mathcal{Y} \circledast \mathcal{X} \right]_{\delta\beta\gamma} \mathbf{Z}_{\alpha\delta}^\top \\
&= \sum_{\delta=1}^{P} \left( \sum_{c=1}^{N} \sum_{h=1}^{K} \sum_{w=1}^{K} \mathcal{Y}_{\delta chw} \mathcal{X}_{c(h+\beta-1)(w+\gamma-1)} \right) \mathbf{Z}_{\alpha\delta}^\top \\
&= \sum_{c=1}^{N} \sum_{h=1}^{K} \sum_{w=1}^{K} \sum_{\delta=1}^{P} \mathcal{Y}_{\delta chw} \mathbf{Z}_{\alpha\delta}^\top \mathcal{X}_{c(h+\beta-1)(w+\gamma-1)}.
\end{aligned} \tag{17}
$$

Therefore,

$$\left(\mathcal{Y} \times_1 \mathbf{Z}^\top\right) \circledast \mathcal{X} = \left(\mathcal{Y} \circledast \mathcal{X}\right) \times_1 \mathbf{Z}^\top.$$

∎

## 6.5 Proof of Equation 7a

**Equation 7a.** $\quad \mathcal{W}_{i+1} \circledast \left(f\left(\mathcal{Y}_i \circledast \mathcal{X}_i\right) \times_1 \mathbf{Z}_i^\top\right) = \left(\mathcal{W}_{i+1} \times_2 \mathbf{Z}_i\right) \circledast f\left(\mathcal{Y}_i \circledast \mathcal{X}_i\right)$

*Proof.* For simple notation, let $\mathcal{W}' = \mathcal{W}_{i+1}, N' = N_{i+1}$ and subscript $i$ is omitted. Also, let $\mathcal{X}' = f(\mathcal{Y} \circledast \mathcal{X})$,

$$
\begin{aligned}
\left[\mathcal{W}' \circledast \left(f\left(\mathcal{Y} \circledast \mathcal{X}\right) \times_1 \mathbf{Z}^\top\right)\right]_{\alpha\beta\gamma} &= \left[\mathcal{W}' \circledast \left(\mathcal{X}' \times_1 \mathbf{Z}^\top\right)\right]_{\alpha\beta\gamma} \\
&= \sum_{c=1}^{N'} \sum_{h=1}^{K} \sum_{w=1}^{K} \mathcal{W}'_{\alpha chw} \left[\mathcal{X}' \times_1 \mathbf{Z}^\top\right]_{c(h+\beta-1)(w+\gamma-1)} \\
&= \sum_{c=1}^{N'} \sum_{h=1}^{K} \sum_{w=1}^{K} \mathcal{W}'_{\alpha chw} \left(\sum_{\delta=1}^{P} \mathcal{X}'_{\delta(h+\beta-1)(w+\gamma-1)} \mathbf{Z}_{c\delta}^\top\right) \\
&= \sum_{c=1}^{N'} \sum_{h=1}^{K} \sum_{w=1}^{K} \sum_{\delta=1}^{P} \mathcal{W}'_{\alpha chw} \mathcal{X}'_{\delta(h+\beta-1)(w+\gamma-1)} \mathbf{Z}_{c\delta}^\top.
\end{aligned}
\tag{18}
$$

$$
\begin{aligned}
\left[\left(\mathcal{W}' \times_2 \mathbf{Z}\right) \circledast f\left(\mathcal{Y} \circledast \mathcal{X}\right)\right]_{\alpha\beta\gamma} &= \left[\left(\mathcal{W}' \times_2 \mathbf{Z}\right) \circledast \mathcal{X}'\right]_{\alpha\beta\gamma} \\
&= \sum_{\delta=1}^{P} \sum_{h=1}^{K} \sum_{w=1}^{K} \left[\mathcal{W}' \times_2 \mathbf{Z}\right]_{\alpha\delta hw} \mathcal{X}'_{\delta(h+\beta-1)(w+\gamma-1)} \\
&= \sum_{\delta=1}^{P} \sum_{h=1}^{K} \sum_{w=1}^{K} \left(\sum_{c=1}^{N'} \mathcal{W}'_{\alpha chw} \mathbf{Z}_{\delta c}\right) \mathcal{X}'_{\delta(h+\beta-1)(w+\gamma-1)} \\
&= \sum_{c=1}^{N'} \sum_{h=1}^{K} \sum_{w=1}^{K} \sum_{\delta=1}^{P} \mathcal{W}'_{\alpha chw} \mathcal{X}'_{\delta(h+\beta-1)(w+\gamma-1)} \mathbf{Z}_{c\delta}^\top.
\end{aligned}
\tag{19}
$$

Therefore,

$$\mathcal{W}' \circledast \left(f\left(\mathcal{Y} \circledast \mathcal{X}\right) \times_1 \mathbf{Z}^\top\right) = \left(\mathcal{W}' \times_2 \mathbf{Z}\right) \circledast f\left(\mathcal{Y} \circledast \mathcal{X}\right).$$

∎

## 6.6 Image Classification Results on ImageNet

In Table 4, we present the test results of VGG-16 and ResNet-34 on ImageNet. We prune only the last convolution layer of VGG-16 as most of the parameters come from fully connected layers. For ResNet-34, we prune all convolution layers in equal proportion. Due to the large scale of the dataset, the initial accuracy right after the pruning drops rapidly as the pruning ratio increases. However, our merging recovers the accuracy in all cases, showing our idea is also effective even for large-scale datasets like ImageNet.

## 6.7 Effect of Hyperparameter $t$

In this section, we analyze the effect of the hyperparameter $t$ in the case of ResNet-56. The average cosine similarity between filters of ResNet is lower than that of over-parameterized models. Thus, it is more sensitive to the hyperparameter $t$, which is used as the minimum cosine similarity threshold of compensated filters.

Fig. 5(a) shows the distribution of the maximum cosine similarity, which is the value between each filter and the nearest one. The variance and the median value of the maximum cosine similarity tend to decrease toward the back layers of ResNet-56. In the back layers, the cosine similarity values

Table 4: Performance comparison of pruning and merging for VGG-16 and ResNet-34 on ImageNet dataset without fine-tuning. 'Param. #' denotes absolute parameter number of pruned/merged models. For VGG, 'Last-{}%' denotes the pruning ratio of the last convolution layer.

| Pruning Ratio | Criterion | Top 1 Acc. | | | Top 5 Acc. | | | Param. # |
|---|---|---|---|---|---|---|---|---|
| | | Prune | Merge | Acc. ↑ | Prune | Merge | Acc. ↑ | |
| **VGG-16** | | 73.36% | | | 91.51% | | | 138M |
| Last-50% | $l_1$-norm | 57.00% | 61.18% | **4.18**% | 81.05% | 84.90% | **3.85**% | |
| | $l_2$-norm | 56.85% | 60.82% | **3.98**% | 81.41% | 84.90% | **3.49**% | 85M |
| | $l_2$-GM | 54.96% | 60.54% | **5.58**% | 79.79% | 84.81% | **5.01**% | |
| Last-60% | $l_1$-norm | 47.70% | 53.78% | **6.08**% | 73.61% | 80.44% | **6.83**% | |
| | $l_2$-norm | 47.99% | 54.13% | **6.13**% | 74.45% | 80.74% | **6.29**% | 55M |
| | $l_2$-GM | 47.23% | 53.12% | **5.88**% | 73.67% | 80.22% | **6.55**% | |
| Last-70% | $l_1$-norm | 34.75% | 43.26% | **8.51**% | 60.06% | 71.62% | **11.56**% | |
| | $l_2$-norm | 35.15% | 43.99% | **8.84**% | 60.86% | 72.49% | **11.63**% | 64M |
| | $l_2$-GM | 36.18% | 43.57% | **7.39**% | 62.04% | 72.18% | **10.14**% | |
| **ResNet-34** | | 73.31% | | | 91.42% | | | 21M |
| 10% | $l_1$-norm | 62.08% | 66.30% | **4.22**% | 84.85% | 87.35% | **2.50**% | |
| | $l_2$-norm | 63.71% | 66.77% | **3.07**% | 85.78% | 87.63% | **1.85**% | 19M |
| | $l_2$-GM | 61.79% | 66.58% | **4.79**% | 84.46% | 87.53% | **3.07**% | |
| 20% | $l_1$-norm | 40.66% | 53.95% | **13.29**% | 67.32% | 78.66% | **11.34**% | |
| | $l_2$-norm | 42.80% | 55.37% | **12.57**% | 68.65% | 79.59% | **10.93**% | 17M |
| | $l_2$-GM | 43.47% | 55.70% | **12.23**% | 69.12% | 79.75% | **10.63**% | |
| 30% | $l_1$-norm | 12.52% | 35.56% | **23.04**% | 29.43% | 61.07% | **31.64**% | |
| | $l_2$-norm | 17.06% | 37.43% | **20.37**% | 35.84% | 62.63% | **26.79**% | 15M |
| | $l_2$-GM | 15.81% | 36.05% | **20.24**% | 33.45% | 61.29% | **27.84**% | |

(a) Maximum cosine similarity distribution

(b) Relationship between $t$ and accuracy

Figure 5: Cosine similarity analysis of ResNet-56 on CIFAR-10. (a) is the boxplot of the maximum cosine similarity of filters within each layer. The $x$-axis represents the position of the layer in the model. For example, '1-1' indicates the first pruned layer in the first residual block. (b) is the relationship between the cosine threshold $t$ and accuracy under different pruning ratios: 30%, 40%, and 50%. $\lambda$ is set to 1 to select filters based on the cosine similarity only.

are mostly distributed between 0.1 and 0.3. This level of cosine similarity might seem too low to be meaningful. Nevertheless, the highest accuracy is obtained when all the filters with the cosine similarity over 0.15 are compensated for, as shown in Fig. 5(b). This trend appears in all three pruning ratios. As the pruning ratio increases, both the accuracy gain and fluctuation are more prominent.