[Reviews · NeurIPS 2020]

Review 1

Summary and Contributions: The paper proposes an idea to compensate the pruned neurons by merging their contributions (approximately) to the retained neurons (actually to the nearest retained one according to cosine similarity) for ReLU networks. This ensures that pruning does not lose too much in accuracy and the pruned network retains most of the performance of the original network without finetuning. Experiments are conducted on fashion mnist and cifar datasets that shows the benefits of the method.

Strengths: 1) The idea of compensating for pruned neurons is interesting and useful in practice. The common drawback of pruning is, it reduces the learned accuracy and one needs to retrain it but this approach partly alleviates the need for retraining in relu networks. 2) Overall the paper is well written and the method is clearly explained.

Weaknesses: Some of the weaknesses/comments in my opinion are as follows: 1) The idea works only with relu please mention it upfront in abstract and introduction. It is not clear how this can be extened to other acitvation functions. Please comment. 2) In Alg. 1, the "scale" simply assinged to z without aggregating to the existing value of z. So it is not clear if two pruned neurons are found to be closest to one retained neuron, how is this situation handled? Please clarify this. 3) Please consider discussing "pruning-at-initialization" methods [a,b] which would have the same training time and complexity as the proposed merging method as both the approaces train the network only once. It would be interesting to see how this approach compares against those ones. 4) The benefits of this method could be deemed limted as this method performs similar to the vanilla pruning method which does not do any merging when finetuning is allowed. But I agree that the method focuses on pruning without fine-tuning. [a] Lee, N., Ajanthan, T. and Torr, P.H., 2019. Snip: Single-shot network pruning based on connection sensitivity. ICLR. [b] Frankle, J. and Carbin, M., 2019. The lottery ticket hypothesis: Finding sparse, trainable neural networks. ICLR. Post rebuttal update: I was confused with the Alg.1 update to z which the authors clarified. I would recommend the authors to improve the clarity in the updated version as well in addition to other promised discussions. I was positive and I retain my rating.

Correctness: The method, claims and experimental setup are correct.

Clarity: The paper is clearly written.

Relation to Prior Work: Most of the works are adequately discussed. Please consider discussing pruning at initialization methods as mentioned in weaknesses.

Reproducibility: Yes

Additional Feedback: Please consider releasing the code for reproducibility.


Review 2

Summary and Contributions: This paper proposes neuron merging, which aims to compress two FC-layers or conv-layers across ReLU layers. The basic idea is that FC-layer before ReLU could be decomposed into two matrix, and one could be merged with FC-layer after ReLU if that matrix satisfy certain conditions. Similar results could be obtained on conv-layers. The Authors provide algorithms for this specific decomposition. The method is verified on a set of network architectures on dataset MNIST, CIFAR-10/100.

Strengths: + The idea looks very interesting, and the derivation sounds correct.

Weaknesses: - If a convolution layer could be decomposed into a lightweight conv-layer plus an 1*1 conv-layer, Corollary-1.1 is more easy to be proved, followed Theorem 1. Interestingly, this property is proved by the other paper "Network Decoupling: From Regular to Depthwise Separable Convolutions" in BMVC 2018. - Most modern CNN networks just have one FC layers. Hence, the merging of conv-layers is more interesting and practical. - Another special case is thus proposed. How to handle network with depth-wise separable layers like MobileNet (v1/v2/v3)? - The network this tested is very limited, and the dataset is also small. How about other more network architectures like DenseNet, and modern NAS searched networks? Are there any results on ImageNet?

Correctness: Some special cases like depthwise convolution are not discussed in the paper.

Clarity: It generally easy to follow, and the illustration is very clear.

Relation to Prior Work: See my weakness points.

Reproducibility: Yes

Additional Feedback: I give the rating based on the current status. However, I basically love the idea. I would like to seriously consider upgrading my rating if the authors could address my concerns in the rebuttal. update after rebuttal: I appreciate the authors providing additional experiments to address my concerns. If this paper is accepted, the authors are strongly suggested to consider the generalization capability of this work.


Review 3

Summary and Contributions: This paper proposes an approach to merge the factor that is generated from pruning the previous layer into the next layer. It proves the condition for which this pruning is exact. Experiment results show that this simple procedure can improve the results of multiple state-of-the-art network pruning algorithms that do not involve fine-tuning.

Strengths: The methodology is simple and sound, and it significantly improves the results of previous algorithms.

Weaknesses: This is not necessarily a weakness, but I was wondering what would happen if a proper matrix decomposition algorithm, such as a non-negative matrix factorization, is applied as the decomposition algorithm in Alg. 1, instead of the current "MostSim" heuristic. It is unclear whether the MostSim heuristic is optimal and authors did not mention that aspect. Given that the improvement was already significant, I think it's OK to take the paper as-is. But the paper would be strengthened if NMF-type algorithms are tried against the MostSim heuristic.

Correctness: I haven't checked line-by-line, but I think the proofs are correct.

Clarity: It's well-written and easy to read.

Relation to Prior Work: Prior work were addressed properly as far as I know. I don't work on the area of pruning hence may miss recent work.

Reproducibility: Yes

Additional Feedback:

[Author Response · NeurIPS 2020]

We thank the reviewers for their positive feedbacks and valuable suggestions. We address their comments below.

**Reviewer #1**

**1. Activation functions**: We will mention that our method is applicable to the ReLU function upfront. Although this work only covers the case of ReLU function for rigorous mathematical proof, we empirically observed that neuron merging can be extended to many activation functions as seen in Table 1. We leave a theoretical explanation of why neuron merging works well with various activation functions as a future work.

**2. Clarification of Alg. 1**: According to Alg. 1, scaling matrix $\mathbf{Z}_i$ has the size of $P_{i+1} \times N_{i+1}$. If two pruned neurons $(a, b)$ share the same retained neuron $(c)$ as the closest one, $scale$ for each pruned neuron will be stored in the corresponding entry in $\mathbf{Z}_i$ (same row $(c)$, different columns $(a, b)$). When $\mathbf{Z}_i$ is merged with the weights of the next layer, this separately stored $scale$ plays the role of compensating for each removed neuron as shown in Fig. 2. The corresponding dimension of a pruned neuron is multiplied by its $scale$ and added to that of the retained neuron.

**3. Comparison with "pruning-at-initialization" methods**: We appreciate the advice, and we will add the discussion in the related works. In short, "pruning-at-initialization" methods have the advantage of less overhead at training time. In contrast, our approach can be adopted even when the model is trained without any consideration of pruning.

**Reviewer #3**

**1. Regarding the proof of Corollary 1.1**: It is an insightful suggestion to use the property proved in the other paper to prove Corollary 1.1. However, we think providing a self-sufficient proof in the current form is not a bad idea, either.

**2. Expansion to complex network architectures**: One simple approach to (approximately) handle depthwise separable layers is to consider the combination of PW and DW layers as one regular convolution layer. Let us assume the layer structure of 1*1 - depthwise - 1*1. Neuron merging aims to reconstruct the output feature map of the second 1*1 conv, even after pruning the filters in the first 1*1 conv and the corresponding channels in the depthwise conv. We can use the outer product of the two as the input of Alg. 1, and set $scale$ as the norm ratio between the two tensors. Preliminary experimental results of MobileNetV1 is shown in Table 2. Further research is needed to expand neuron merging to more complex architectures such as NAS-searched networks or DenseNet.

**3. Results on ImageNet**: In Table 3, we present the test results of VGG-16 and ResNet-34 on ImageNet. We prune only the last convolution layer of VGG-16 as most of the parameters come from fully connected layers. For ResNet, we prune all convolution layers in equal proportion. Due to the large scale of the dataset, the initial accuracy right after the pruning drops rapidly as the pruning ratio increases. However, our merging recovers the accuracy in all cases, showing our idea is also effective even for large-scale datasets like ImageNet.

**Reviewer #4**

**1. Considering other matrix decomposition algorithms (i.e., NMF)**: As far as we know, there is no NMF-type algorithm that can satisfy the conditions in Thm. 1. Also, NMF-type algorithms have difficulty in handling minus values frequent in the weight matrix. Nonetheless, we tried to decompose the weight using NMF algorithm by zeroizing negative weights. For LeNet-5 on Fashion-MNIST, NMF showed an inferior accuracy of 15.80% on average when "MostSim" algorithm showed 88.69%. In the case of 3D/4D tensor, NMF is not easily applicable due to the mismatch of tensor shape. Having said that, we agree that "MostSim" heuristic is not optimal and plan to search for better decomposition methods.

| Activation | Baseline | Prune | Merge | Acc.↑ |
|---|---|---|---|---|
| Tanh | 90.72% | 67.81% | 81.32% | **13.51%** |
| SoftSign | 91.14% | 76.98% | 87.18% | **10.20%** |
| ELU | 91.25% | 77.73% | 89.68% | **11.95%** |
| SELU | 90.47% | 25.17% | 80.28% | **55.11%** |
| LeakyReLU | 93.89% | 92.10% | 93.46% | **1.36%** |
| Hardswish | 91.93% | 88.30% | 91.72% | **3.42%** |

Table 1: Comparison of pruning and merging with various activation functions (VGG-16 on CIFAR-10). Pruning strategy is the same as the original paper.

| Pruning Ratio | Prune | Merge | Acc. ↑ | Param. # |
|---|---|---|---|---|
| Baseline | | 87.90% | | 3.2M |
| 40% | 84.52% | 85.84% | **1.32%** | 1.4M |
| 50% | 77.34% | 80.74% | **3.40%** | 1.0M |
| 60% | 39.39% | 55.88% | **16.49%** | 0.7M |

Table 2: Performance comparison of pruning and merging for MobileNetV1 on CIFAR-10. We prune all layers in equal proportion.

| Pruning Ratio | Top 1 Acc. | | | Top 5 Acc. | | | Param. # |
|---|---|---|---|---|---|---|---|
| | Prune | Merge | Acc. ↑ | Prune | Merge | Acc. ↑ | |
| **VGG-16** | | 73.36% | | | 91.51% | | 138M |
| Last-50% | 57.00% | 61.18% | **4.18%** | 81.05% | 84.90% | **3.85%** | 85M |
| Last-60% | 47.70% | 53.78% | **6.08%** | 73.61% | 80.44% | **6.83%** | 75M |
| Last-70% | 34.75% | 43.26% | **8.51%** | 60.06% | 71.62% | **11.56%** | 64M |
| **ResNet-34** | | 73.31% | | | 91.42% | | 21M |
| 10% | 62.08% | 66.30% | **4.22%** | 84.85% | 87.35% | **2.50%** | 19M |
| 20% | 40.66% | 53.95% | **13.29%** | 67.32% | 78.66% | **11.34%** | 17M |
| 30% | 12.52% | 35.56% | **23.04%** | 29.43% | 61.07% | **31.64%** | 15M |

Table 3: Performance comparison of pruning and merging for VGG-16 and ResNet-34 on ImageNet. $l_1$-norm is used as the pruning criterion.

[Meta-Review · NeurIPS 2020]

Two of three reviews are favorable. The reviewers point out that the theoretical contribution, a method for pruning (merging) without requiring fine-tuning, is solid. The paper contains sufficient supporting empirical evidence for the efficacy of the method. The paper is therefore accepted.